# Long-Term Results of Pediatric Congenital Pulmonary Malformation: A Population-Based Matched Case–Control Study with a Mean 7-Year Follow-Up

**DOI:** 10.3390/children10010071

**Published:** 2022-12-29

**Authors:** Susanna Nuutinen, Eveliina Ronkainen, Marja Perhomaa, Terttu Harju, Juha-Jaakko Sinikumpu, Willy Serlo, Teija Dunder

**Affiliations:** 1Research Unit of Clinical Medicine, Medical Research Center (MRC) Oulu, Oulu University Hospital, University of Oulu, 90220 Oulu, Finland; 2Division of Pediatric Surgery, Department of Children and Adolescents, Oulu University Hospital, University of Oulu, 90220 Oulu, Finland; 3Division of Neonatal Medicine, Department of Children and Adolescents, Oulu University Hospital, University of Oulu, 90220 Oulu, Finland; 4Division of Pediatric Radiology, Department of Radiology, Oulu University Hospital, University of Oulu, 90220 Oulu, Finland; 5Research Unit of Medical Imaging, Physics and Technology (MIPT), Oulu University Hospital, University of Oulu, 90220 Oulu, Finland; 6Research Unit of Internal Medicine, Medical Research Center (MRC) Oulu, Oulu University Hospital, University of Oulu, 90220 Oulu, Finland; 7Pulmonary Unit, Department of Medicine, Oulu University Hospital, University of Oulu, 90220 Oulu, Finland; 8Division of Allergology and Pulmonology, Department of Children and Adolescents, Oulu University Hospital, University of Oulu, 90220 Oulu, Finland

**Keywords:** pulmonary malformation, long-term outcome, congenital, thoracotomy

## Abstract

Symptomatic congenital pulmonary malformations (CPMs) are a group of anomalies involving the lungs. The long-term outcomes of these patients are not well known. The present research aimed to study the pulmonary function, respiratory morbidity, and health-related quality of life (QoL) of patients treated for CPMs. All children (<16 years of age) treated for CPMs in 2002–2012 (in Oulu University Hospital) were invited to the follow-up visit. Altogether, there were 22 patients, out of which 17 (77%) participated. The mean follow-up time was 6.6 (ranged from 3 to 16) years. Pulmonary function tests, diffusing capacity, respiratory morbidity, and QoL were determined as the primary outcomes. Potential residual malformations and lung anatomy were investigated using computer tomography (CT) imaging. The outcomes were compared to the age- and sex-matched healthy controls. The forced expiratory volume at 1 s (FEV_1_, Z-score) remained lower in operated patients compared to the healthy controls (−1.57 ± SD 1.35 vs. −0.39 ± SD −0.86, *p*-value 0.005). There were no differences in respiratory morbidity or QoL between the patients and the controls. The surgical approach (lobectomy vs. partial resection) did not affect lung function. A younger age (<1 year of age) at the time of the surgery seemed to result in a higher lung capacity, but the finding was not statistically significant. Patients with CPMs treated with surgery were satisfied with their wellbeing in the long-term. A lower lung function did not have an impact on their wellbeing. However, there was a slight decrease in lung function compared to the healthy controls, and a clinical follow-up of the patients was recommended.

## 1. Introduction

Congenital pulmonary malformations (CPMs) are a group of various anomalies involving the lungs [1]. Due to the increasing availability of prenatal ultrasound globally and improvements in diagnostics, the incidence of these malformations has increased, present in approximately 1 to 2500 live births [2,3].

There is no consensus about the treatment of CPMs among symptomless patients [4,5,6]. The surgical treatment for CPMs includes a partial resection or lobectomy. Mini-invasive procedures may be associated with a decreased need for secondary operations compared to open surgery [7,8,9]. Early surgery is thought to be useful, given that infants’ lungs present a greater capacity to grow after surgery compared to older children [10]. Fetal surgery has also been suggested for extreme cases [2], but it is not a standard. Other treatment options include thoracoamniotic shunting [11] in microcystic CPMs with oedemic fetuses, maternal use of corticosteroids in cases of microcystic CPM and oedemic fetuses [12], and watchful waiting.

There is no agreement regarding the clinical or radiographic long-term results of patients with CPMs. In general, fetal hydrops, large malformations, mediastinal shifts, or cardiac insufficiency predict poor outcomes [11].

One previous study compared the operative treatment early and later in childhood, and found no differences in pulmonary function [13]. In another study, patients who underwent thoracoscopic lobectomy were compared to healthy controls, and no differences were found in pulmonary function [8]. Thoracoscopy and thoracotomy patients were compared regarding shoulder mobility, scapular winging, and other thoracic muscular deformities [14]. They found differences in scapular winging in favor of thoracoscopy. To the best of our knowledge, the quality of life in these patients has not been studied previously.

There is no understanding of the optimal frequency and length of the follow-up schemas in patients with asymptomatic CPMs [5,15]. However, if left untreated, there is a higher risk of lung infection throughout life, and complicated surgeries may be needed at a later stage [16]. There may also be a higher risk of malignant transformations [17].

The aim of this study was to report on the pulmonary function, respiratory morbidity, and health-related quality of life of patients treated for pulmonary malformations in their childhood. Another aim was to study the effect of age at the time of surgery on long-term outcomes.

## 2. Materials and Methods

### 2.1. Study Design

This was a population-based age- and sex-matched case–control study in the area of Oulu University Hospital, Oulu, Finland. The study cohort included only patients with congenital pulmonary malformations (CPMs) treated during 2002–2012. All children aged at least five years at follow-up were invited to participate (n = 22), of which 16 participated. Patients were identified from hospital registries using the diagnosis code Q33 in the International Classification of Diseases (ICD-10). The age at the time of operation, the type of operation, and the possible complications were collected from the medical records. Age- (±6 months) and sex-matched control cases were randomly selected from the population register (Digital and Population Data Services Agency). In total, 20 controls for each patient were invited, and the first volunteer was enrolled for an evaluation. The participants and their controls did not differ in height of weight. There was no difference in passive smoke exposure between the cases and the controls. Finally, there were 32 individuals in total who comprised the study cohort. The mean follow-up time was 6.6 years (ranging from 3 to 16 years).

The clinical evaluation included an investigation of spine deformities using scoliometry. This measures spine rotation, for which more than 6 degrees is considered to indicate scoliosis. Thoracic, thoracolumbar, and lumbar spine scoliometries were measured separately. The range of movements of the shoulders (abduction, frontal elevation, and rotations) and upper body flexibility in means of lateral bending were measured. The symmetry of the thoracic cage and the scar after thoracotomy were visually estimated. All clinical measures and evaluations were performed by the primary author with 23 years of experience in pediatric surgery (SN). The patients also reported on their subjective evaluations of the cosmetic and functional results.

### 2.2. Lung Function, Diffusing Capacity, and Respiratory Morbidity

The assessment of lung function included spirometry and exercise tests for bronchial hyper-responsiveness, as well as the diffusion capacity of the lungs for carbon monoxide (DLCO). Spirometry was performed using computer-based equipment. Flow–volume curves were obtained to determine the forced vital capacity (FVC), forced expiratory volume in 1 s (FEV_1_ = the maximum amount of air that the subject could forcibly expel during the first second following maximal inhalation), the FEV_1_/FVC proportion, and the maximum expiratory flow at 50% of FVC (MEF_50_). A bronchodilation test was performed. Postbronchodilator spirometry was carried out 15 min after the administration of 400 µg salbutamol via a spacer. An increase of at least 12% in FEV_1_ was considered to be a significant bronchial reversibility [18,19]. The DLCO was measured using a single-breath technique; the patient would first breathe normal resting breaths, followed by a full exhalation. Then, the patient would rapidly inhale the test gas. The DLCO, total lung capacity (TLC), residual volume (RV), and RV/TLC were recorded using a Jaeger MasterScreen^®^ PFT device (Viasys Healthcare GmbH, Hoechberg, Germany). All measurements of the respiratory system were performed in accordance with the American Thoracic Society (ATS) and European Respiratory Society (ERS) guidelines [18,19]. The data on lung function were standardized for height, age, and sex. The results of spirometry were reported on using global, multiethnic GLI-2012 regression equations (Z-score) [20].

An exercise test was used to assess exercise-induced bronchoconstriction (EIB). The ERS Task Force guidelines were applied in standardizing the settings of free-run tests [21]. A decline of ≥15% on FEV_1_ was considered EIB.

The children’s’ history of respiratory diseases was inquired from the parents using the validated Finnish version of the International Study of Asthma and Allergies in Childhood questionnaire (ISAAC) [22].

### 2.3. Imaging

Computer tomography (CT) of the lungs was carried out for all study patients, but not for the healthy controls, due to radiation exposure. The potential residual malformations, tumor tissues, overinflation of a lung, scarring, atelectasis, and other bronchial abnormalities were analyzed. All images were reviewed by a pediatric radiologist with 18 years of experience in pediatric imaging (MP).

### 2.4. Health-Related Quality of Life

Health-related quality of life (HRQoL) was assessed using the Pediatric Quality of Life inventory (PedsQL) version 4.0 [23].

### 2.5. Statistics

Descriptives of the study population were presented using frequencies, proportions, and M-values (mean, median) with a standard deviation (SD) and range, when appropriate. Comparisons between the groups were performed by using a paired sample *t*-test for continuous variables and a chi-square test for categorical values. An exact test was used for small (<5) groups. A *p*-value of <0.05 was considered statistically significant. All *p*-values were two-tailed. All analyses were performed with IBM using SPSS version 26 for Windows (SPSS Inc. Chicago, IL, USA).

## 3. Results

### 3.1. Patient Characteristics

There were 11 boys and 6 girls among these cases. All malformations had been prenatally diagnosed and none were prenatally treated. In total, 16 out of 17 children (94%) were operatively treated, and those 16 were included in this study. Nine patients were operated on at <1 year of age. The malformations included one intralobar sequestration in the left upper lobe, two extralobar sequestrations in the left side, four congenital pulmonary malformations (CPAMs) type one located in the right side (two in the upper lobe and two in the lower lobe), and seven CPAMs type two located on both sides in the lower lobes except for one that was located in the left upper lobe. These malformations’ volumes were one lobe or less.

### 3.2. Clinical Findings

The majority of the patients presented a slight axial rotation on scoliometry (2–4 degrees) but did not reach the diagnostic threshold of scoliosis. Radiographs were required for one patient, but no intervention was needed for their scoliosis. There was no difference in the spine posture between the patients and the controls (Table 1). The inclination of the spine did not follow the side of the thoracotomy; instead, it varied.

### 3.3. Lung Function, Diffusing Capacity, and Respiratory Morbidity

The thoracic shape was asymmetric in three patients (18%) and one (6%) control case (*p* = 0.301). These findings were the result of a visual evaluation, but were not visible on CT. No differences in shoulder mobility were found between the patients and the controls. Neither group had any differences found in the upper body flexibility (lateral bending).

Regarding subjective outcomes, one patient reported a minor cosmetic disadvantage from the scar (6%).

FEV_1_ was significantly lower in the CPM patients than in the healthy controls (*p* = 0.008) (Table 2). There were no significant differences in other lung function tests, diffusing capacity, exercise tests, or bronchodilation tests (Table 3). Furthermore, no differences were found in the prevalence of asthma, pulmonary medication, wheezing, cough, or pulmonary infections between the patients and the controls (Table 3).

The lung function tests of patients operated on before one year of age, after one year of age, and the healthy controls were compared, but there were no significant differences between the groups. This study did not find a difference between those who had undergone pulmonary lobectomy and those who had sparing resection (data not shown).

### 3.4. Imaging Findings

One patient, treated with lung sparing surgery for a congenital pulmonary malformation (CPAM) type two, presented residual lung cysts in a small area less than 2 cm in diameter. No other residual malformations or tumors were found among the patients. Eight patients (47%) showed overinflation (Figure 1), and all but one had minor scarring (Figure 2, black arrow). One patient with CPAM type two, treated with lung sparing surgery, had air trapping, and another had bronchial thickening. No bronchiectasis, detectable deformities of the thorax, or rib fusions were found.

### 3.5. Health-Related Quality of Life

There were no differences in HRQoL between the patients and their healthy controls (Table 4).

## 4. Discussion

The main finding of this comprehensive, population-based, case–control study of congenital lung malformations was that the long-term (>5 years) outcomes were excellent. There were no major differences in any lung function tests, except that FEV_1_ was lower in CPM patients. The clinical importance of a decreased FEV_1_ remains unclear, since it did not increase morbidity or affect the physical functioning of the patients.

Thoracotomy in pediatric patients is known to have impact on the posture of the spine, but seldom to the point, the treatment is needed [24]. All our patients had posterolateral thoracotomy and there were no differences in thoracic spine deformities or other musculoskeletal sequelae. In a previous study, musculoskeletal outcomes were compared with thoracoscopy and thoracotomy, and more winging of the scapula was found in thoracotomy patients, but no other differences [14,25]. The findings in the present study suggested that thoracotomy is well tolerated and does not cause significant sequelae in the long-term.

Furthermore, the operative treatment was successful in all cases but one, meaning that close to all patients were primarily treated successfully. The only patient with a residual cyst was treated with a sparing resection.

Previous studies tried to find out if the patients’ age at the operation or the operative approach (sparing surgery vs. lobectomy) is meaningful to the outcome [15], but no significant differences have been found thus far. The present study did not give additional value to this question, but the possibility of residual malformations was more likely with sparing surgery [9,14,26]. In this study, overinflation was a common finding in CT, even with patients operated on at an early age, suggesting that alveolar dilatation plays a bigger role in filling the thoracic space postoperatively than previously estimated [27]. There was one lobar emphysema that disappeared during the follow-up (this patient was excluded from the analysis), and this could justify avoiding interventions for at least some symptomless patients.

It is noteworthy that the quality of life was equal in all categories between the patients and healthy controls.

The main limitation of our study was its small study population. Due to the small study population, slight differences between groups may not have been found. The heterogeneity of the CPMs was also a limitation of this study. Different CPM types might need different kinds of approaches regarding the time and type of surgery or follow-up. There might have been a selection bias in participation, while those who participated could have been more symptomatic and, thus, prone to participate (Table 1). Closer particulars of nonparticipants were not available, and participants and nonparticipants could not be compared. Another limitation was the lack of CT in the control cases. Nevertheless, as an ionizing investigation, it was abandoned for ethical reasons.

The strength of this study was its unique study design; age- and sex-matched control cases were selected from the national population register. They did not differ in weight or height. To the author’s knowledge, this was a novel research describing the long-term clinical, radiographic and quality of life outcomes of CPMs, compared to normal controls. Furthermore, the patients and controls were examined by one experienced surgeon, and all images were analyzed by one experienced pediatric radiologist.

## 5. Conclusions

Congenital pulmonary malformations, most usually treated operatively in early childhood, did not decrease the quality of life or increase morbidity during childhood compared to normal controls. The resection of the lungs decreased FEV_1_, but the clinical importance of this finding was unclear.

## Figures and Tables

**Figure 1 children-10-00071-f001:**
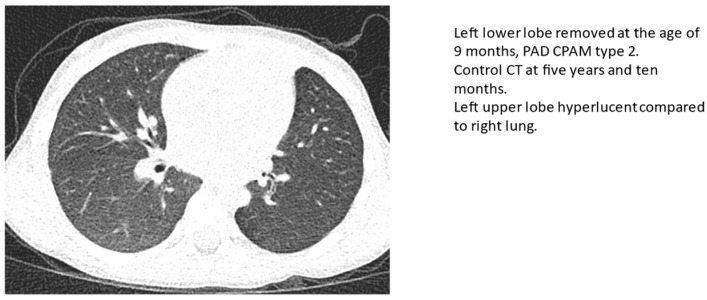
Overinflation in operated lung.

**Figure 2 children-10-00071-f002:**
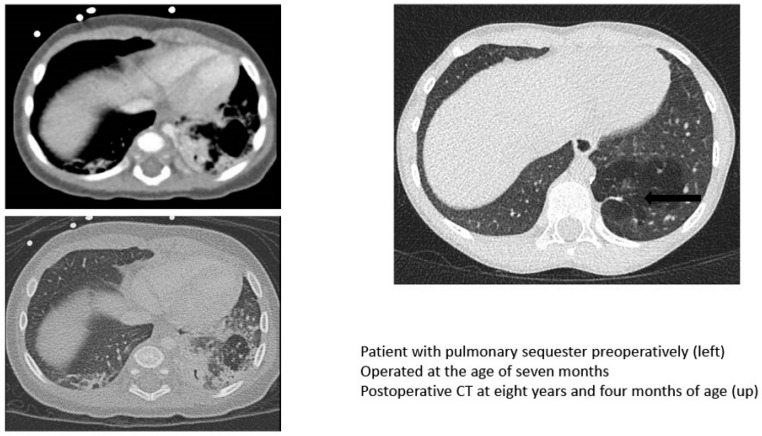
CT images preoperative and showing postoperative scarring and overinflation (arrow).

**Table 1 children-10-00071-t001:** Results from the scoliometry evaluation.

	Patient	Control	*p*-Value
Thoracal spine, inclination to the right, degrees	0.9 ± 1.5	1.1 ± 1.5	0.750
Thoracal spine, inclination to the left, degrees	1.6 ± 2.2	1.1 ± 1.3	0.545
Thoracolumbar spine, inclination to the right	0.9 ± 1.0	1.0 ± 1.2	0.843
Thoracolumbar spine, inclination to the left	2.0 ± 2.6	0.7 ± 1.3	0.174
Lumbar spine, inclination to the right	1.0 ± 1.0	1.3 ± 1.6	0.655
Lumbar spine, inclination to the left	1.4 ± 1.1	0.8 ± 1.1	0.258

**Table 2 children-10-00071-t002:** Comparison of lung function in patients and controls.

Lung Function Variable	Patients	Controls	*p*-Value
FVC ^1^, Z-score	−0.65 ± 3.10	0.08 ± 1.38	0.397
FEV_1_ ^2^, Z-score	−1.52 ± 1.38	−0.35 ± 0.87	0.008
FEV_1_/FVC, Z-score	−0.71 ± 1.63	−0.52 ± 1.20	0.700
MEF_50_ ^3^, Z-score	−1.42 ± 1.88	−0.96 ± 0.95	0.387
Change in FEV_1_ after exercise test, %	5.4 ± 16.2	−1.5 ± 12.0	0.195
Decrease in FEV_1_ > 15% after exercise test, n (%)	0	1 (6.3%)	0.294
Change in FEV_1_ after bronchodilator, %	4.5 ± 4.6	0.5 ± 11.7	0.212
Increase in FEV_1_ ≥ 12% after bronchodilator, n (%)	2 (17.6%)	2 (17.6%)	1.000
DLCO ^4^ mmol/(minkPa)	6.5 ± 2.6	7.4 ± 2.8	0.432
TLC ^5^	3.6 ± 1.7	4.2 ± 2.1	0.435
RV ^6^	1.5 ± 0.7	1.6 ± 0.9	0.833
RV/TLC percentage	43 ± 12	37 ± 13	0.285
RV/TLC percentage of reference value	204 ± 49	162 ± 66	0.203

Data were resented in mean ± SD unless otherwise specified. ^1^ FVC, forced vital capacity; ^2^ FEV_1_, forced expiratory volume in 1 s; ^3^ MEF_50_, maximum expiratory flow at 50% of FVC; ^4^ DLCO, diffusion capacity of the lungs for carbon monoxide; ^5^ TLC, total lung capacity; ^6^ RV, residual volume.

**Table 3 children-10-00071-t003:** Baseline characteristics and respiratory morbidity of CPM (congenital pulmonary malformation) patients and healthy controls.

Follow-Up Visit	Patients	Controls	*p*-Value
Subjects, n	16	16	
Male, n (%)	10 (62.5%)	10 (62.5%)	
Female, n (%)	6 (37.5%)	6 (37.5%)	
Height, cm	138.2 ± 24.0	135.0 ± 23.5	0.705
Weight, kg	35.7 ± 20.7	33.7 ± 15.9	0.771
BMI ^1^	17.2 ± 3.4	17.5 ± 2.5	0.795
Age at follow-up visit, years	9.3 ± 3.8	9.4 ± 3.9	0.924
Range, years	5.4–17.0	5.3–17.5	
Ever had asthma	3 (18.8%)	1 (6.3%)	0.300
Asthma medication ^2^ during the past year	2 (12.5%)	2 (12.5%)	0.154
Wheezing during respiratory tract infection	7 (43.8%)	8 (50%)	0.886
Wheezing during the past year	5 (31.3%)	3 (50%)	0.886
Doctor’s appointment because of wheezing	4 (25%)	8 (50%)	0.095
Dry cough at night without infection	3 (18.8%)	3 (18.8%)	1.000
Respiratory symptoms during exercise	1 (6.3%)	2 (12.5%)	0.330
Prolonged cough or sputum	0 (0%)	1 (6.3%)	0.325
Pneumonias during last year diagnosed by doctor	1 (6.3%)	1 (6.3%)	1.000
History of lung surgery in the family	1 (6.3%)	0 (0%)	0.325

Data were presented as mean ± SD or n (percentage). ^1^ BMI, body mass index; ^2^ regular or periodical inhaled corticosteroid treatment.

**Table 4 children-10-00071-t004:** Quality of life questionnaire.

	Patient	Control	*p*-Value
Child Self-Report			
Physical functioning	89.5 ± 11.8	88.3 ± 13.2	0.793
Emotional functioning	80.6 ± 18.0	79.7 ± 15.5	0.876
Social functioning	91.6 ± 11.0	88.4 ± 12.0	0.451
School functioning	78.8 ± 17.6	83.1 ± 13.6	0.438
Physical health summary score ^1^	89.5	88.3	
Psychosocial health summary score ^2^	83.7	83.7	
Total scale score ^3^	85.1 ± 11.5	84.8 ± 10.3	0.940
Parent Proxy Report			
Physical functioning	85.9 ± 13.2	87.9 ± 15.7	0.706
Emotional functioning	77.1 ± 19.4	79.4 ± 17.0	0.737
Social functioning	85.9 ± 18.2	82.8 ± 20.7	0.654
School functioning	81.3 ± 16.6	83.8 ± 15.5	0.663
Physical health summary score ^1^	85.9	84.8	
Psychosocial health summary score ^2^	81.4	82	
Total scale score ^3^	82.9 ± 13.3	83.5 ± 13.8	0.907

^1^ The physical health summary score was the same as the physical functioning scale score. ^2^ The psychosocial health summary score mean was computed as the sum of the items divided by the number of items answered in the emotional, social, and school functioning scales. ^3^ The total scale score mean was computed as the sum of all items divided by the number of items answered on all scales. Higher scores indicated a better health-related quality of life. Data were presented as means ± SDs.

## Data Availability

Data available on request from the authors.

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
