# Peer review of "Long-Term Results of Pediatric Congenital Pulmonary Malformation: A Population-Based Matched Case–Control Study with a Mean 7-Year Follow-Up"

_children, 2022, doi:10.3390/children10010071_

Round 1

Reviewer 1 Report

This study showed the long-term outcomes of congenital pulmonary malformation (CPM). The authors found there were no significant differences other than FEV1 compared to the normal healthy children in terms of clinical presentation and quality of life. It would be worth publishing their findings after revision regarding the below.

They didn't need to describe any non-participants since they did not PARTICIPATE in this study. Comparison between participants and no-participants was irrelevant in Table 1.

Since they studied deformity of the thorax, they should have excluded a patient without surgery. Besides, this patient showed spontaneous regression of lung lesions, in which case he(or she) became a normal child during the follow-up period. The authors should exclude this patient from the comparison to the normal cohort.

A table showed result 3.2  would be helpful to see the comparison of deformity of the thorax and spine between patients and control groups.

In figure 2, an arrow showing a scar would be helpful.

Author Response

They didn't need to describe any non-participants since they did not PARTICIPATE in this study. Comparison between participants and no-participants was irrelevant in Table 1.

We thank the reviewer for the excellent remarks. According to instructions we removed table 1. as irrelevant content.

Since they studied deformity of the thorax, they should have excluded a patient without surgery. Besides, this patient showed spontaneous regression of lung lesions, in which case he(or she) became a normal child during the follow-up period. The authors should exclude this patient from the comparison to the normal cohort.

We also removed the one patient conservatively treated from the analysis as well pointed and did the analysis again without that patient or his/her control.

A table showed result 3.2  would be helpful to see the comparison of deformity of the thorax and spine between patients and control groups.

This is a fine remark, and we added table 1. to give information of the scoliometry evaluation.

In figure 2, an arrow showing a scar would be helpful.

This is good point and we have added an arrow.

Reviewer 2 Report

This is a study presenting long term follow up after surgical treatment of 17 patients with CPM. The authors are assessing the consequences over the lung function, the clinical aspect and the quality of life of children with CPM. Their conclusion is that there are no differences in terms of respiratory morbidity and quality of life in these patients.  There are a few studies on this matter but, there is still debate over the follow up protocol, long term results, even for the best treatment options (timing, operative vs non-operative).

Line 89: Does the control group match the CPM group in terms of age, sex, weight, height? Please specify.  

 Line 131: CT is an invasive procedure. What was the reason to perform Lung CT for all the patient? Was there a medical justification?   

Line 148, Patients’ characteristics:

·        The anatomo-pathological characteristics of the CPM (type, size, precise location), are in direct relation with the surgical procedure, timing, etc

·        The type of surgery (the extent of lung resection and remaining lung tissue) might be in relation with the long term outcomes

·        These characteristics should be accurately specified and analysed.

Line 152: What kind of sequestration, intralobar or extralobar?

Line 218: Since there is no statistical significance, it is dangerous to make this statement. Moreover, the data for this comparison has to be clearly presented in the results section in order to conduct discussion over them. Please rephrase.

Line 226: This might be an important finding of the study and it worst discussing it more. One of the criticism for open vs thoracoscopic approach for CPM surgery is that the scar is bigger and it might induce thoracic and spine deformities.

Line 233: The heterogeneity of the CPM is also a limitation of the study. The different types of the CPM have different clinical consequences and might require different therapeutic approach meaning type and timing for surgery, follow up, etc.

Overall, the discussion section must be improved. There are a few very good recent publications that deal with this matter and might be referred to (Hall et al, Seminar is Pediatric Surgery 2017, Tivnan et al. Radiologic clinics of North America, 2022, etc)

Line 261, ethical approval: A number and date of approval shall be provided.

Line 265: References are not as per journal style

Author Response

We thank the reviewer for the valuable remarks. 

Line 89: Does the control group match the CPM group in terms of age, sex, weight, height? Please specify. 

This is a good point made and we added the information to the text.

Line 131: CT is an invasive procedure. What was the reason to perform Lung CT for all the patient? Was there a medical justification?

This is an important and necessary question. We dicussed this matter when planning our study protocol. We consulted our pediatric radiologist of the matter. The new computer tomography set-up has quite low radiation load. For thoracic ct it equals the amount of 2 months of backround radiation for a child of 20 kilos and 4 months of backround radiation for a child of 40 kilos. As there was a clinical interest of possible residual malformation, we thought this was acceptable.

Line 148, Patients’ characteristics:

  • The anatomo-pathological characteristics of the CPM (type, size, precise location), are in direct relation with the surgical procedure, timing, etc
  • The type of surgery (the extent of lung resection and remaining lung tissue) might be in relation with the long term outcomes
  • These characteristics should be accurately specified and analysed.

This is a valuable remark and we have added this information to the text.

Line 152: What kind of sequestration, intralobar or extralobar?

This was a good point and we specified that in the text.

Line 218: Since there is no statistical significance, it is dangerous to make this statement. Moreover, the data for this comparison has to be clearly presented in the results section in order to conduct discussion over them. Please rephrase.

This is a valuable remark. This paragraph has been partly removed and rephrased.

Line 226: This might be an important finding of the study and it worst discussing it more. One of the criticism for open vs thoracoscopic approach for CPM surgery is that the scar is bigger and it might induce thoracic and spine deformities.

We thank the reviewer for this comment. We have discussed this more in the discussion part.

Line 233: The heterogeneity of the CPM is also a limitation of the study. The different types of the CPM have different clinical consequences and might require different therapeutic approach meaning type and timing for surgery, follow up, etc

This is a valuable addition to the limitations. We have added this to the text.

Overall, the discussion section must be improved. There are a few very good recent publications that deal with this matter and might be referred to (Hall et al, Seminar is Pediatric Surgery 2017, Tivnan et al. Radiologic clinics of North America, 2022, etc)

We thank for this comment. We have revised the discussion section and believe it is much better now.

Line 261, ethical approval: A number and date of approval shall be provided.

This information is added.

Line 265: References are not as per journal style

We have corrected the reference style.